# *Streptomyces* sp. AN090126 as a Biocontrol Agent against Bacterial and Fungal Plant Diseases

**DOI:** 10.3390/microorganisms10040791

**Published:** 2022-04-08

**Authors:** Khanh Duy Le, Nan Hee Yu, Ae Ran Park, Dong-Jin Park, Chang-Jin Kim, Jin-Cheol Kim

**Affiliations:** 1Department of Agricultural Chemistry, Institute of Environmentally Friendly Agriculture, College of Agriculture and Life Sciences, Chonam National University, Gwangju 61186, Korea; khanhld2387@gmail.com (K.D.L.); nanheeyu0707@gmail.com (N.H.Y.); arpark9@naver.com (A.R.P.); 2Institute of New Technology, Academy of Military Science and Technology, 17 Hoangsam, Caugiay, Hanoi 100000, Vietnam; 3Industrial Bio-Materials Research Center, Korea Research Institute of Bioscience and Biotechnology, 125 Gwahak-ro, Yuseong-gu, Daejeon 34141, Korea; dongjin@kribb.re.kr (D.-J.P.); changjin@kribb.re.kr (C.-J.K.)

**Keywords:** *Streptomyces*, antimicrobial activity, biocontrol agent, VOCs, synergistic effect

## Abstract

Bacteria and fungi are major phytopathogens which substantially affect global agricultural productivity. In the present study, *Streptomyces* sp. AN090126, isolated from agricultural suppressive soil in Korea, showed broad-spectrum antagonistic activity against various phytopathogenic bacteria and fungi. In the 96-well plate assay, the fermentation filtrate of *Streptomyces* sp. AN090126 exhibited antimicrobial activity, with a minimum inhibitory concentration (MIC) of 0.63–10% for bacteria and 0.63–3.3% for fungi. The MIC of the partially purified fraction was 20.82–250 µg/mL for bacteria and 15.6–83.33 µg/mL for fungi. Gas chromatography–mass spectrometry (GC-MS) analysis revealed that AN090126 produced various volatile organic compounds (VOCs), including dimethyl sulfide and trimethyl sulfide, which inhibited the growth of pathogenic bacteria and fungi in in vitro VOC assays. In pot experiments, the fermentation broth of *Streptomyces* sp. AN090126 reduced tomato bacterial wilt caused by *Ralstonia solanacearum*, red pepper leaf spot caused by *Xanthomonas euvesicatoria*, and creeping bentgrass dollar spot caused by *Sclerotinia homoeocarpa* in a dose-dependent manner. Moreover, the secondary metabolites derived from this strain showed a synergistic effect with streptomycin sulfate against streptomycin-resistant *Pectobacterium carotovorum* subsp. *carotovorum*, the causative agent of Kimchi cabbage soft rot, in both in vitro and in vivo experiments. Therefore, *Streptomyces* sp. AN090126 is a potential biocontrol agent in controlling plant diseases caused by pathogenic bacteria and fungi, specifically by the streptomycin-resistant strains.

## 1. Introduction

Global agricultural productivity is affected by a wide array of fungal, bacterial, viral, and nematode diseases. Fungi accounted for 42% and bacteria for 27% of the 14% global crop production loss caused by plant diseases, and this damage can worsen under climate change [1,2]. In light of this, the use of pesticides for crop protection has increased, ultimately increasing the risk of chemical contamination of the food chain and ecosystems. To address this problem, biological control using microbial antagonists has been proposed as an alternative to chemical. Biological control is more resilient to the development of pest resistance than the conventional chemical and has advantages such as minimum or no residual toxicity and environmental pollution [3].

Plant disease control by microbial antagonists has garnered much attention during the past few decades. Microbial antagonists are living microorganisms used for controlling plant diseases [4]. Biopesticides derived from microbial antagonists such as bacteria, fungi, and viruses, among others, represent 5% of the crop protection market, and bacteria represent 90% of all such agents [5]. To date, *Bacillus* spp., *Pseudomonas* spp., and *Streptomyces* spp. have been reported as effective biocontrol agents in controlling diverse phytopathogens [6,7,8]. Numerous recent studies have focused on the use of *Streptomyces* spp. as biocontrol agents to control various plant diseases caused by bacteria and fungi [9,10,11,12,13,14]. By several mechanisms, bacteria of the genus *Streptomyces* can effectively suppress various plant fungal and bacterial diseases. The production of volatile and non-volatile antibiotics, cell wall-degrading enzymes, hyperparasitism on pathogenic organisms, promotion of plant growth, and induction of systemic resistance in the host plant have been reported as their action mechanisms [15,16,17].

Since 1955, after its discovery and isolation from *Streptomyces griseus* subsp. *griseus*, the aminoglycoside antibiotic streptomycin has been applied in plant agriculture. It is commonly applied to control plant bacterial diseases such as fire blight caused by *Erwinia amylovora*, soft rot caused by *Pectobacterium* spp., leaf spot caused by *Xanthomonas campestris* pv. *vesicatoria*, and crown gall caused by *Agrobacterium tumefaciens*. However, the emergence of resistant bacterial strains has impeded the control of plant bacterial diseases using streptomycin. Since the discovery of the first resistant strain in Florida in 1960, streptomycin-resistant strains of several bacteria, including *E. amylovora*, *Pectobacterium carotovorum*, *Xanthomonas* spp., and *Pseudomonas* spp., have been detected [18]. In this context, new antibiotics must be developed that can effectively suppress the development of plant diseases caused by streptomycin-resistant bacterial pathogens. Meanwhile, a secondary method to overcome streptomycin resistance is the use of streptomycin in combination with a material that produces a synergistic effect against antibiotic-resistant pathogens [19,20,21]. Although a number of secondary metabolites have been identified in *Streptomyces* spp. [22], to the best of our best knowledge, there is no report on metabolites producing synergistic effects with the antibiotics used in agriculture.

In this work, we discovered that *Streptomyces* sp. AN090126 exhibited potent antibacterial and antifungal activities as well as synergistic activity with streptomycin. Hence, the objectives of the present study were: (1) To identify and characterize *Streptomyces* sp. AN090126. (2) To examine the antibacterial and antifungal activities of the FB and PPF in vitro. (3) To evaluate the disease control efficacy of the FB in controlling various plant diseases caused by bacteria and fungi. (4) To explore the interactions of the secondary metabolites produced by *Streptomyces* sp. AN090126 with streptomycin against streptomycin-resistant bacteria.

## 2. Materials and Methods

### 2.1. Culture and Fermentation Conditions

*Streptomyces* sp. AN090126 was isolated from agricultural soil at the Korea Research Institute of Bioscience and Biotechnology and maintained on the International *Streptomyces* Project 2 (ISP2; Becton, Dickinson and Company, Sparks, MD, USA) agar medium at 30 °C for 7 days. The stock culture was prepared in 30% glycerin (Sigma-Aldrich; St. Louis, MO, USA) and kept at −80 °C for long-term storage. For the fermentation of antibacterial metabolites, a single colony was inoculated into 5 mL of sterile ISP2 broth medium in a test tube and then incubated at 30 °C and 180 rpm for 2 days to obtain the seed culture. The 1% seed culture was transferred into 100 mL of GSS medium [14] in a 500 mL Erlenmeyer flask and incubated as described above for 7 days. The fermentation broth (FB) was filtered via a membrane (0.2 µm; ADVANTEC, Tokyo, Japan) for the antimicrobial test.

### 2.2. Phytopathogenic Bacteria and Fungi

Antagonistic activity of the fermentation filtrate (FF) and the partially purified fraction (PPF) of *Streptomyces* sp. AN090126 was assayed against 16 phytopathogenic bacteria including two *Acidovorax* strains (*A. avenae* subsp. *cattleyae* and *A. konjaci*), *A. tumefaciens*, *E. amylovora*, *Burkholderia glumae*, *Claviabacter michiganensis* subsp. *michiganensis*, three *Pectobcaterium* strains (*P. carotovorum* subsp. *carotovorum*, *P. chrysanthemi*, and Sr-Pcc), two *Pseudomonas* strains (*P. syringae* pv. *actinidiae* and *P. syringae* pv. *lachryman*), *R. solanacearum*, and four strains of *Xanthomonas* spp. (*X. euvesicatoria*, *X. arboricola* pv. *pruni*, *X. axonopodis* pv. *citri*, and *X. oryzae* pv. *oryzae*). The bacteria were obtained from Rural Development Administration, Dong-A University, South Korea, and Suncheon National University, South Korea. They were maintained on tryptic soy agar (TSA; Becton, Dickinson and Company, Sparks, MD, USA) medium at 30 °C from stock and cultured on tryptic soy broth (TSB; Becton, Dickinson and Company) medium at 30 °C. Ten phytopathogenic fungi were also used in this study. *Botrytis cinerea*, *Colletotrichum coccodes*, *Fusarium oxysporum* f. sp. *lycopersici*, *Magnaporthe oryzae*, and *Rhizoctonia solani* were supplied by the Korea Research Institute of Chemical Technology. *Cryphonectria parasitica*, *Raffaelea quercus-mongolicae*, and *S. homoeocarpa* were provided by the Korea Forest Research Institute. *Fusarium graminearum* and *Pythium cactorum* were kindly supplied by the Seoul National University and Korea Agricultural Culture Collection, respectively. The fungi were cultured on potato dextrose agar (PDA; Becton, Dickinson and Company, Sparks, MD, USA) and potato dextrose broth (PDB; Becton, Dickinson and Company, Sparks, MD, USA) media and incubated at 25 °C. All pathogens were stocked in 30% glycerin at −80 °C for long time.

### 2.3. Identification and Morphological and Biochemical Observations of the AN090126 Strain

The isolated ribosomal RNA fragment was amplified by PCR using 518f (5’-GTATTACCGCGGCTGCTGG-3’), 785f (5’-GTGGACTACCAGGGTATCTA-3’), and 805r (3’-GACTACCAGGGTATCTAATC-5’). The PCR amplification was performed through initial denaturation at 95 °C for 5 min, then 30 cycles of 95 °C for 30 s, 55 °C for 30 s, 72 °C for 90 s, and a final elongation at 72 °C for 10 min. The obtained sequence was subjected to the BLAST tool to compare with the reference sequences of *Streptomyces* available in GenBank. A phylogenetic tree was constructed by Molecilar Evolution Genetics Analysis (MEGA) software in version 10.2 using the neighbor-joining method [23] with 1000 bootstrap replicates. Whole gene sequences were also uploaded and performed pairwise genome by the Orthologous Average Nucleotide Identity (OrthoANI, EzBiocloud, Seoul, South Korea) tool to calculate OrthoANI values [24]. On the other hand, colony morphology of the AN090126 strain was studied under a scanning electron microscope on ISP medium. Enzyme productivity, and the ability to grow in different temperatures, range of pH, and concentrations of NaCl (Sigma-Aldrich, St. Louis, MO, USA) on medium were conducted as described previously [25]. FB was filtered and subjected on high performance liquid chromatography (1260 Infinity/G4212B; Agilent Co., Santa Clara, CA, USA) to analyze the production of organic acids. 

### 2.4. Preparation of PPF

The FB (100 mL) of *Streptomyces* sp. AN090126 was acidified to a pH of 3.6 with 5 M hydrochloric acid and then centrifuged to remove the precipitate. The supernatant was filtered through filter paper (Whatnam No.2; Sigma-Aldrich, St. Louis, MO, USA) and precipitated in ice-cold acetone (FB:acetone, 1:4). The precipitate was collected by centrifugation at 4000 rpm and 4 °C for 15 min and then washed with 80% ice-cold acetone. The obtained precipitate was dissolved in water and subjected to two-step ultrafiltration. The solution was successively filtered through 10 and 3 kDa centrifugal filters (Milipore, Carrigtwahill, Ireland). The flow-through was collected and then loaded onto the Sephadex G-10 (Sigma-Aldrich, St. Louis, MO, USA) columns equilibrated with water. The active fractions were combined and lyophilized to obtain 84.8 mg of PPF. The antibacterial activity of PPF was assessed by bioautography against the test bacteria.

### 2.5. Minimum Inhibitory Concentration (MIC) Assay

The MICs of the FF and PPF were measured against 16 bacterial and 10 fungal plant pathogens using the broth dilution method, as described previously for bacteria [26] and fungi [27]. Briefly, 10% of the FB and 250 µg/mL of PPF were diluted 2-fold. An aliquot of the bacterial (10^5^ CFU/mL in TSB) or fungal mycelial (1% in PDB) suspension was added to each well to obtain the final volume of 100 µL. Both streptomycin sulfate (Ss) (Sigma-Aldrich) and oxolinic acid (OA) (Sigma-Aldrich) were used as the positive controls. The plates were incubated for 1 to 2 days at 30 °C for bacteria and for 3 to 7 days at 25 °C for fungi. The MIC value was the lowest concentration of the sample that completely inhibited the growth of the test microorganisms. The experiment consisted of three replicates, and the entire experiment was repeated twice. 

### 2.6. Checkerboard Assay

This assay was performed with the PPF and streptomycin sulfate against bacteria that are resistant to streptomycin sulfate, and the fractional inhibitory concentration (FIC) was determined as described previously [28]. Two-fold diluted PPF in the range of 0.125–2 times the MIC value was added along the x-axis and streptomycin sulfate was added along the y-axis of a 96-well plate. The bacteria (10^5^ CFU/mL in TSB) were inoculated and then incubated for 1 to 2 days at 30 °C. The fractional inhibitory concentration indices (FICIs) of all combinations were determined using the following formulas:FIC_A_ = (MIC of A in combination) ÷ (MIC of A alone)
FIC_B_ = (MIC of B in combination) ÷ (MIC of B alone)
FICI = FIC_A_ + FIC_B_(1)

The FICIs were interpreted as follows; ≤0.5, synergy; 0.5 to ≤0.75, partial synergy; 0.75 to ≤1, additive; 1 to ≤4, indifferent; and >4, antagonism. The experiment was repeated twice with three replicates.

### 2.7. Antimicrobial Bioassays of VOCs

An I-plate system was designed to examine the effect of AN090126’s VOCs on the growth of plant pathogenic bacteria and fungi [29,30]. Briefly, a 50 µL-aliquot of AN090126 spore suspension (10^5^ CFU/mL) was spread on one side containing the ISP-2 agar medium 3 days before inoculation, except in control plates. Then, the test microorganisms were inoculated at 5 points on the other side containing the TSA medium with 2 µl of each bacterial suspension (10^6^ CFU/mL) per point, or the test fungi were inoculated on the other side containing the PDA medium with a mycelial agar plug. The plate was sealed twice with parafilm and placed in an incubator at 28 °C for 72 h for bacteria or 3 to 7 days for fungi. After incubation, the test bacterial colonies were harvested with sterile distilled water and the bacterial growth was measured based on OD at 600 nm using a UV spectrophotometer (Shimazu UV-1601 Spectrophotometer; Shimadzu Co., Kyoto, Japan). For antifungal activity, the colony area was calculated using the following formula:Colony area (mm^2^) = [π × (a × b)] ÷ 4(2)
where a is the length of the fungal colony (mm) and b is the width of the fungal colony (mm).

### 2.8. Collection and Analysis of Volatile Organic Compound (VOC) Production Using Solid-Phase Microextraction (SPME)/Gas Chromatography-Mass Spectrometry (GC-MS)

The VOCs were collected using SPME with a polydimethylsiloxane/divinylbenzene (65 µm) fiber (Supelco Inc., Bellefonte, PA, USA) in triplicate at 50 °C for 30 min [30]. The VOCs were analyzed using the Shimadzu GCMS-QP2010 gas chromatograph (70 eV; Shimadzu Co., Kyoto, Japan) equipped with a capillary column of DB-5MS (30 m × 0.25 mm, film thickness 0.25 µm; Agilent Technologies, Inc., Santa Clara, CA, USA). The carrier gas was helium with a flow rate of 1 mL/min. The oven temperature of GC analysis was programmed as 50 °C for 2 min at the beginning, followed by an increase to 250 °C at 10 °C/min, and held for 20 min. The mass spectrometer was performed in the positive electron ionization mode at 70 eV at 200 °C and scan range of 50–400 *m/z*. The mass spectra of VOCs were compared with available data in the WILEY8 Library to identify these compounds.

### 2.9. Plant Materials

Tomato (Seokwnag cultivar; Farm Hannong Co., Ltd, Seoul, South Korea), red pepper (Josaengsintopgochu cultivar; Nongwoobio Co., Ltd., Suwon, South Korea), and Kimchi cabbage (Chunkwang cultivar; Sakada Korea, Seoul, South Korea) were sown in plastic pots (diameter, 6.0 cm) containing nursery soil and grown with 12 h of light per day in an incubation room. The plants were transplanted into bigger plastic pots (diameter, 7.5 cm) 24 h before treatment. Creeping bentgrass was grown in a plastic pot (diameter, 7.5 cm) at 25 °C for 4 weeks, and the seedlings were cut at a height of 20 mm from the mouth of the pots 1 day before inoculation.

### 2.10. In Vivo Assay

The biological control efficacy of *Streptomyces* sp. AN090126 was evaluated against the causative agents of tomato bacterial wilt, red pepper bacterial leaf spot, Kimchi cabbage bacterial soft rot, and creeping bentgrass dollar spot. These assays were performed as described previously [26,27]. The FB was diluted 5-, 10-, and 20-fold with distilled water, and 250 µg/mL of tween 20 was added to each dilution. The disease control efficacy of the combination of the FB (10- or 20-fold dilution) and streptomycin sulfate (200 µg/mL) was evaluated against Kimchi cabbage soft rot caused by Sr-Pcc. The positive controls included bactericide Ilpum (1000-fold dilution) (oxolinic acid 20% WP; Dongbang Agro, Seoul, South Korea) and fungicide Horikua (2000-fold dilution) (Tebucolazole 25% WP; Farm Hannong Co., Seoul, South Korea). The negative control was tween 20 solution (250 µg/mL). The in vivo assays were described in detail in the Appendix A.

### 2.11. Statistical Analysis

The data of the experiments were assessed with one-way analysis of variance (ANOVA) using SPSS version 23.0 (SPSS Inc., Chicago, IL, USA). MIC values were assessed by Duncan’s multiple-range test, and FICI values were assessed with Tukey’s honestly significant difference (HSD) test as a function of the microbial test (factor), the sample treated (covariable), and their interaction. Significant differences among treatments in in vivo experiments were evaluated with Fisher’s least significant difference test. A *p*-value of ≤0.05 was considered to be statistically significant.

## 3. Results

### 3.1. Identification of Streptomyces sp. AN090126

The strain presented the typical morphology of a Gram-positive, non-motile, filamentous, greenish-gray, and aerobic bacterium. Scanning electron microscopy revealed that the aerial hyphae were non-fragmented and most spores were rod-shaped, with a fragmented spore chain and rough surface (Figure 1A). The substrate mycelium was slender and greenish-beige, and it produced a beige-brown water-soluble pigment. Additionally, physiological tests showed that the strain could grow on a medium containing up to 7% sodium chloride at a temperature range of 28–45 °C (optimal temperature, 30 °C) and a pH range of 4–10 (optimum pH, 7). Biochemical tests revealed that the strain could produce cellulase, lipase, and protease, but not amylase (Appendix A). Moreover, it produced numerous organic acids, including oxalic acid, citric acid, tartaric acid, and succinic acid, among others (Appendix A). To identify the strain, a phylogenetic tree was constructed based on the 16S rRNA (1474 bp) gene sequence. Based on the 16S rRNA comparison using the BLAST tool, AN090126 was closely related to *Streptomyces mauvecolor* NBRC 13,854^T^ with similarity values of 99.31%, and a further analysis by the neighbor-joining tree also confirmed that AN090126 belongs to the genus *Streptomyces* (Figure 1B). The 16S rRNA gene sequence was deposited in GenBank under the accession number MT982961. The obtained OrthoANI value was lower than 80%, which represented a different genome species with other members of *Streptomyces* spp. (Appendix A). The data indicated that the AN090126 strain represents a novel species of genus *Streptomyces*.

### 3.2. Antimicrobial Activity of Streptomyces sp. AN090126

*Streptomyces* sp. AN090126 showed broad-spectrum antimicrobial activity against 16 plant pathogenic bacteria and 9 phytopathogenic fungi. Both FF and PPF showed antibacterial activity against the test bacteria (Table 1). *A. avenae* subsp. *cattleyae* and *A. konjaci* were the most sensitive to the fermentatin broth and PPF, followed by *X. oryzae* pv. *oryzae* and *R. solanacearum*. The FF and PPF moderately inhibited the cell growth of *E.*
*amylovora*, *P.*
*carototvora* subsp. *carototvora*, *P. chrysanthemi*, *P. syringae* pv. *actinidiae*, *P. syringae* pv. *lachrymans*, *X. euvesicatoria*, *X. arboricola* pv. *pruni*, and *X. axonopodis* pv. *citri,* as well as of the Sr-Pcc strain. *A.*
*tumefaciens*, *B. glumae*, and *C. michiganensis* subsp. *michiganensis* were relatively insensitive to both samples compared to the other bacteria.

Furthermore, both the fermentation broth and PPF exhibited remarkable antifungal activity against the mycelial growth of the test fungi (Table 2), with the MIC ranging from 0.63–3.33% for the former and 15.6–83.33 µg/mL for the latter. *B. cinerea* was the most sensitive, while *C. coccodes* was relatively insensitive. Interestingly, the mycelial growth of *M. oryzae*, the causal agent of rice blast, was slightly inhibited by either the fermentation filtrate or PPF even at high concentrations of 10% and 500 µg/mL, respectively.

### 3.3. VOCs Produced by Streptomyces sp. AN090126 and Their Effects on Microbial Growth

VOCs produced by the AN090126 strain grown on ISP-2 medium were analyzed by headspace SPME/GC–MS after 7 days of incubation. A total of 11 VOCs including alcohols, aromatics, sulfides, and esters were detected (Appendix A). Among these, 2-propyl furan, 3-methyl-1-butanol, 2-methyl-1-butanol, and dimethyl disulfide were the major compounds, followed by 1-octanol, 2,4-dimethyl-hexanoic acid methyl ester, 1-hexanol, 2-ethyl-1-hexanol, dimethyl trisulfide, 1,4-dimethyl benzene, and 1-methoxy-2-methyl benzene (Table 3).

The VOCs produced by the AN090126 strain exhibited inhibitory activity against the cell growth of all phytopathogenic bacteria (Figure 2A). *A. avenae* subsp. *cattleyae* was the most sensitive, with an inhibition rate of approximately 75%; the inhibition rate for the remaining bacteria ranged from 13% to 47%. Furthermore, the VOCs produced by the AN090126 strain suppressed the mycelial growth of the phytopathogenic fungi (Figure 2B). *F. graminearum* was the most sensitive, followed by *C. coccodes* and the remaining fungi. *M. oryzae* was the most insensitive to all VOCs. Microscopy revealed that the hyphae of Fg exposed to the VOCs were thinner and exhibited a slower growth than those of the untreated controls (Appendix A).

### 3.4. Efficacy of the Fermentation Broth of Streptomyces sp. AN090126 in Controlling Plant Bacterial and Fungal Diseases

To investigate the disease control efficacy of FB of *Streptomyces* sp. AN090126 against tomato bacterial wilt caused by *R. solanacearum* and red pepper leaf spot caused by *X. euvesicatoria*, serial dilutions of the FB were applied to 5-week-old tomato and red pepper plants via soil drenching and foliage spraying, respectively. The FB suppressed the development of both bacterial diseases in a dose-dependent manner (Figure 3A,B). Compared with the positive control, the 5-fold dilution showed a comparable disease control efficacy against tomato bacterial wilt (*p* = 0.169) but a significantly lower efficacy against red pepper leaf spot (*p* = 0.035). The highest disease control was 79.24% (F_3.8_ = 19.61, *p* < 0.001) for tomato bacterial wilt and 58.82% (F_3.8_ = 35.05, *p* < 0.001) for red pepper bacterial leaf spot at the 5-fold dilution. Additionally, the FB also suppressed the disease symptoms of dollar spot on the creeping bentgrass in a dose-dependent manner. Disease severity in the creeping bentgrass exposed to the 5-fold diluted FB was reduced by 82.45% (F_3.8_ = 15.92, *p* < 0.001) compared to that in the untreated bentgrass, but was comparable to that in the bentgrass exposed to tebuconazole (*p* = 0.512) (Figure 3C).

### 3.5. Interaction between PPF and Streptomycin Sulfate

#### 3.5.1. In Vitro Experiment

PPF showed a stronger antibacterial activity against the phytopathogenic bacteria that were moderately or highly resistant to streptomycin sulfate, such as *A. avenae* subsp. *cattleyae*, *A. tumefaciens*, *P. chrysanthemi*, *X. axonopodis* pv. *citri*, and Sr-Pcc (Table 1). To examine the interaction between the PPF and streptomycin sulfate, the checkerboard assay was performed against five phytopathogenic bacteria. The combination of the PPF and streptomycin sulfate at several different ratios showed additive or partial synergistic effects against the test bacteria (Table 4). The combination of the PPF and streptomycin sulfate at a ratio of 2:1 showed the strongest synergistic effect against *P. chrysanthemi*, with a fractional inhibitory concentration index (FICI) of 0.58 (*p* < 0.05).

#### 3.5.2. In Vivo Experiment

In in vitro experiments, the combination of the FB of *Streptomyces* sp. AN090126 and streptomycin sulfate showed partial synergistic effects against Sr-Pcc. To examine the disease control efficacy against Kimchi cabbage soft rot caused by Sr-Pcc, the FB of AN090126 and streptomycin sulfate, either alone or in combination, were applied to Kimchi cabbage plants. While oxolinic acid showed very high disease control efficacy, streptomycin sulfate was virtually inactive. The combination of the FB and streptomycin sulfate showed a higher disease control value than the sum of the control values of the individual treatments (Figure 3D). The combination of 10-fold diluted FB and 200 µg/mL streptomycin sulfate showed the highest disease control efficacy, with a control value of 78.48% (F_5.12_ = 26.03, *p* < 0.001), which was comparable to that of oxolinic acid (*p* = 0.864). Similarly, the control value of the combination of the 20-fold diluted FB and 200 µg/mL streptomycin sulfate (60.87%) was significantly higher than the sum of the control values of the individual treatments (*p* < 0.05).

## 4. Discussion

*Streptomyces*—the largest genus of actinobacteria—produce numerous biological substances, which have important applications in human health and agriculture. These substances can suppress pathogen growth via multiple mechanisms, such as the action of antibiotics and wall-degrading enzymes or the induction of host resistance [14,15]. *Streptomyces* spp. are important biocontrol agents of plant diseases as their products represent a major portion of all antibiotics used in agriculture. Interestingly, when the FB of AN090126 strain with chloroform, ethyl acetate, or butanol was partitioned, antimicrobial activity was not detected in any organic solvent layers, but only the aqueous layer exhibited both antibacterial and antifungal activities. This indicated that the active substances are likely to be highly polar compounds, such as peptide or protein. For testing these hypotheses, an ultrafiltration process revealed that molecular sizes of active substances are smaller than 3 kDa, meaning that they do not seem to be proteins. Additionally, their antimicrobial activity was stable at room temperature, similar to that under UV light and low pH (Appendix A). Nevertheless, following exposure to high temperatures of 80 °C and 100 °C for 1 h, the activity decreased by 50% and 80%, respectively. Furthermore, the PPF interacted with ninhydrin in the bioautography assay (Appendix A), indicating that active substances may harbor amino groups [31]. Taken together, the active compounds produced by AN090126 may be hydrophilic peptides that were photo- and acid-stable, but not thermostable. On the other hand, the AN090126 strain displayed the ability to produce hydrolytic enzymes such as cellulase, chitinase, protease, and gelatinase, some of which were reported to play a role in controlling plant disease [15,21,32,33]. The antibacterial and antifungal activities of *Streptomyces* sp. AN090126 were quite similar to those of *Streptomyces* sp. JCK-6131 [14], which also displayed the two biological activities. However, most *Streptomyces* strains have been reported to have either only antibacterial [34,35] or antifungal activity [36,37,38]. Thereby, the AN090126 strain is a potent candidate for developing a biocontrol agent in controlling plant diseases caused by both bacteria and fungi. 

The production of antimicrobial VOCs is one of the biocontrol mechanisms of *Streptomyces* species against plant diseases. Until now, many studies have shown that the VOCs produced by *Streptomyces* can inhibit the growth of plant pathogens. *Streptomyces philanthi* RM-1-138 produced 36 VOCs which inhibit the growth of four plant pathogenic fungi [39]. *S. coeruleoprunus* UAE1 produced antifungal VOCs against *F. solani* [40]. Moreover, *Streptomyces alboflavus* TD-1 produced VOCs that suppressed the mycelial growth of *Fusarium moniloforme*, *Penicillum citrinum*, and three *Aspergillus* spp. [10]. Similarly, in the present study, the VOCs produced by *Streptomyces* sp. AN090126 could inhibit the growth of 9 fungi and 14 bacteria in vitro. Among the test pathogens, *A. avenae* subsp. *cattleyae* and *F. graminearum* were the most sensitive to VOCs. *Streptomyces* sp. AN090126 produced 11 VOCs, which are also produced by other antagonistic *Streptomyces* spp. [39,41]. Among the 11 VOCs, dimethyl disulfide and dimethyl trisulfide were reported to have antimicrobial activity [42,43]. Both compounds completely inhibited the spore germination of *Penicillium italicum* at concentrations of 100 and 1000 µL/L [44] and significantly inhibited the mycelial growth of *R. solani* and *Pythium ultimum* [43]. In another study, dimethyl disulfide, a major VOC produced by *Serrratia plymuthica* IC1270, strongly suppressed the cell growth of *X.*
*euvesicatoria* and *Agrobacterium vitis*, which are the causal agents of crown gall in tomato and grapevine, respectively [42]. The growth of *Serratia marcescens* P87 and *E. coli* WA321 were suppressed by dimethyl trisulfide [45]. In addition, the colony morphology and pigmentation of *S. marcescens* P87 were affected 4 days after exposure to dimethyl trisulfide. In the present study, we observed similar effects. The colony morphology of the exposed bacteria appeared abnormal, and the exposed fungi developed a thinner hyphae and lighter pigmentation than the untreated controls 3 days after incubation. This may be related to the inhibition of quorum sensing, which has been reported in previous studies [46,47]. The remaining nine VOCs produced by *Streptomyces* sp. AN090126 were also reported to be the components of antimicrobial substances produced by *Streptomyces* spp. [39,41].

The bacterial wilt of tomato and leaf spot of red pepper are the major devastating plant diseases worldwide, which lead to substantial economic losses. These diseases cause extensive crop damage and ultimately, significant yield loss [48,49]. To date, many control strategies against these two diseases have been applied, such as the use of resistant varieties, physical-chemical control, and biological control with the main one being biocontrol agents [1,50]. Biocontrol agents are mainly bacteria and their products, which have advantages such as a long-term effect, minimum, or no residual toxicity in the agro-product and environment. Hence, they are gradually replacing synthetic fungicides in controlling a number of plant diseases [3,48,51]. For example, *Bacillus amyloliquefaciens* QL-5 and QL-18, used in combination with an organic fertilizer, showed a biocontrol efficacy of 17% to 87% against the bacterial wilt of tomato under field conditions [52]. *Streptomyces* sp. NEAU-HV9 and its fermentation product (actinomycin D) suppressed the disease symptoms of bacterial wilt in tomatoes with a biocontrol efficacy of 82% and 100%, respectively [53]. In other studies, the disease severity of bacterial leaf spot on red pepper was reduced by 62–67% after treatment with *Bacillus* strains by approximately 71% after treatment with *Streptomyces avermectinius* NBRC14893, and by 66.67% following treatment with the FB of *Paenibacillus elgii* JCK-5075 [6,26]. Similar to our findings, the development of both tomato bacterial wilt and red pepper leaf spot was effectively reduced by the application of *Streptomyces* sp. AN090126.

Dollar spot, which is a common disease for golf of turfgrass, is prevalent from early Spring through to late Fall. The disease can spread out to another place by the mycelium of causal fungus [54]. To control this disease, chemical fungicides have been widely used. Nevertheless, to reduce the use of synthetic fungicides, many studies have focused on the discovery of potent biological agents, such as *Pseudomonas* spp., *Paenibacillus* spp., and *Enterobacter* spp., which can effectively suppress this disease of turfgrass [7,27,55]. In the present study, the FB of *Streptomyces* sp. AN090126 not only showed a broad spectrum in vitro antifungal activity, but also effectively suppressed the development of dollar spots on creeping bentgrass in in vivo assays, and no phytotoxic symptom was observed. From what can be observed, *Streptomyces* sp. AN090126 plays a role as a potential biocontrol agent for bacterial and fungal plant diseases.

Streptomycin inhibits the growth of bacteria by binding with the small ribosomal subunit during protein synthesis. Bacteria carrying specific mutations in the 16S rRNA gene can survive even at high concentrations of streptomycin [56]. In the present study, the secondary metabolites derived from *Streptomyces* sp. AN090126 exhibited a synergistic effect with streptomycin sulfate against streptomycin-resistant bacteria. This result indicates that the mechanism of action of antibacterial metabolites derived from AN090126 is different from that of streptomycin sulfate. Synergistic effects of streptomycin with other antibiotics such as penicillin, cefotaxime, and tetracycline have also been reported [20,57]. The control efficacy of the combination of streptomycin sulfate and AN090126 fermentation was evaluated against Kimchi cabbage soft rot caused by Sr-Pcc. The disease control efficacy of the combination was better than the combined efficacy of individual treatments. Consequently, *Streptomyces* sp. AN090126 can combine with antibiotics to effectively control the plant diseases caused by streptomycin-resistant bacteria.

## 5. Conclusions

In the present study, we propose *Streptomyces* sp. AN090126 as a potential biocontrol agent for the control of plant diseases caused by bacteria and fungi. *Streptomyces* sp. AN090126 produced water-soluble secondary antimicrobial metabolites and VOCs with broad-spectrum antimicrobial activity. The water-soluble metabolites could produce synergistic effects with streptomycin sulfate both in vitro and in vivo. In order to develop novel bactericides and fungicides using *Streptomyces* sp. AN090126, further studies are warranted on the identification of the water-soluble antimicrobial metabolites, development of optimum fermentation and production processes, creation of optimum formulations, assessment of toxicity, and evaluation of disease control efficacy under diverse field conditions.

## Figures and Tables

**Figure 1 microorganisms-10-00791-f001:**
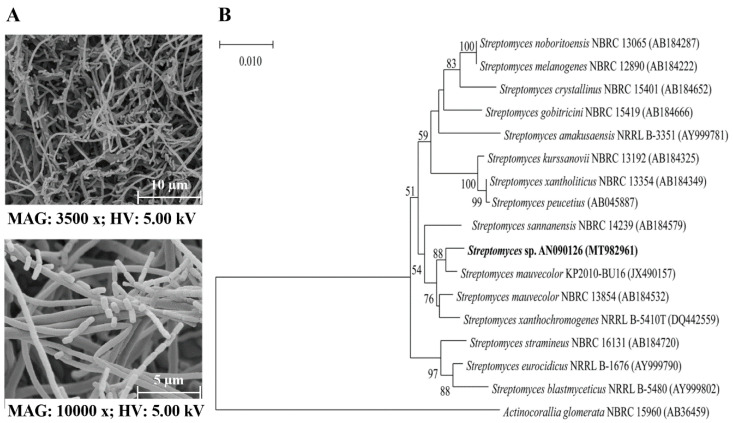
Taxonomy of *Streptomyces* sp. AN090126 based on phylogenetic and morphological characteristics. Scanning electron micrographs (3500× and 10,000×) of the spore morphology of *Streptomyces* sp. AN090126 in IPS medium at 30 °C for 7 days, which showed the aerial hyphae of AN090126 strain were non-fragmented and most spores were rod-shaped with a fragmented spore chain and rough surface (**A**). The phylogenetic relationships between *Streptomyces* sp. AN090126 (MT982961; 1474 bp) and other members of genus *Streptomyces* based on 16S rRNA gene sequences (**B**). The tree was constructed with the neighbor-joining method using the Jukes–Cantor model. The number on the branch points are bootstrap values from 1000 replications. Accession numbers are given in the parentheses. *Micromonospora inyonensis* DSM 46123^T^ was used in the outgroup comparison. The scale bar of 0.010 substitutions per site.

**Figure 2 microorganisms-10-00791-f002:**
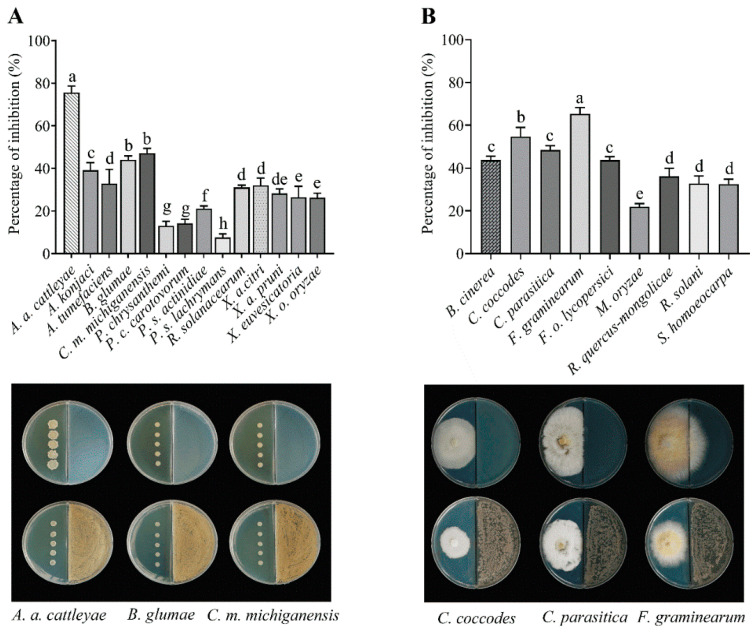
Volatile organic compounds (VOCs) produced by *Streptomyces* sp. AN090126 inhibited the cell growth of plant pathogenic bacteria (**A**) and the mycelial growth of fungi (**B**) in bioassays of the VOCs. The assay was conducted on the I-plate system; the left part was inoculated by pathogens and the right part was *Streptomyces* sp. AN090126. Data are presented as the mean ± standard deviation of three independent triplicates. Means denoted by different letters are considered statistically different (*p* < 0.05) according to Duncan’s multiple range test. VOCs, volatile organic compounds; *A. a. cattleyae*, *A. avenae* subsp. *cattleyae*; *C. m. michiganensis*, *C. michiganensis* subsp. *michiganensis*; *P. c. carotovorum*, *P. carotovorum* subsp. *carotovorum*; *P.s. actinidiae*, *P. syringae* pv. *actinidiae*, *P. s. lachrymans*, *P. syringae* pv. *lachrymans*; *X. a. citri*, *X. arboricola* pv. *citri*; *X. a. pruni*, *X. axonopodis* pv. *citri*, *X. o. oryzae*, *X. oryzae* pv. *oryzae; F. o. lycopersici*, *F. oxysporum* f. sp. *lycopersici*.

**Figure 3 microorganisms-10-00791-f003:**
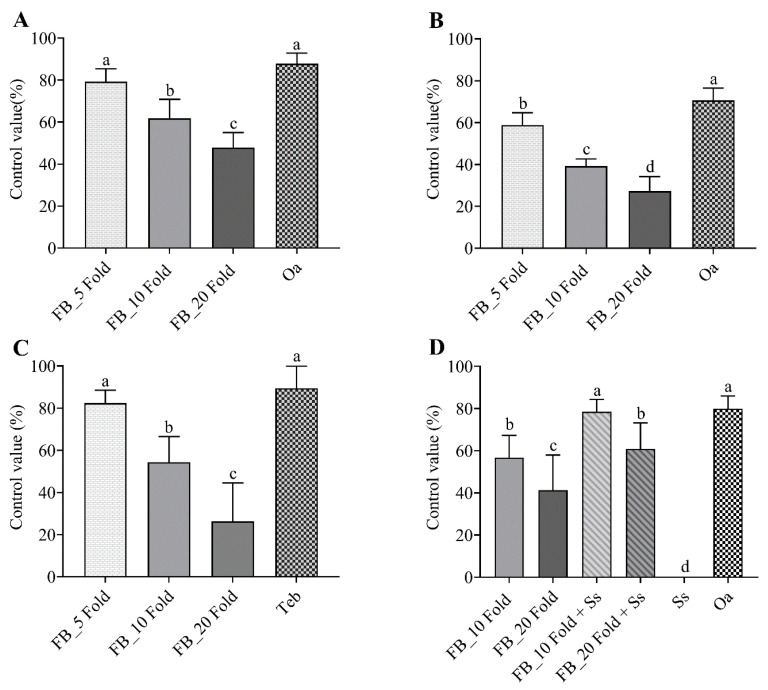
Biocontrol efficacy of the fermentation broth (FB) of *Streptomyces* sp. AN090126 against bacterial wilt of tomato caused by *R. solanacearum* (**A**), leaf spot of red pepper caused by *X. euvesicatoria* (**B**), dollar spot of creeping bentgrass caused by *S. homoeocarpa* (**C**), and soft rot of Kimchi cabbage caused by Sr-Pcc (**D**). In vivo assays were performed by FB of *Streptomyces* sp. AN090126 using the soil drenching method for controlling tomato bacterial wilt, creeping bentgrass dollar spot and Kimchi cabbage soft rot, and foliar spray for red pepper leaf spot diseases. Two bactericides (200 µg/mL of Oa and 200 µg/mL of Ss) or one fungicide (tebucolazole, Teb; 125 µg/mL) were used as positive control(s). FB_5 Fold, 10 Fold and 20 Fold = 5, 10, and 20-fold dilutions of the FB of *Streptomyces* sp. AN090126. Values are presented as the mean ± standard deviation of three runs with triplicates. Bars with the same letters are not significantly different between treatments (*p* < 0.05) followed by Fisher’s least significant difference test.

**Table 1 microorganisms-10-00791-t001:** In vitro assay of antibacterial activity of the fermentation filtrate (FF) and the partially purified fraction (PPF) of *Streptomyces* sp. AN090126 that compared with commercial products against various plant pathogenic bacteria.

Plant Pathogenic Bacteria	Minimum Inhibitory Concentration ^1^			
	FF, %	PPF, µg/mL	Ss ^2^, µg/mL	Oa ^3^, µg/mL
*Acidovorax avenae* subsp. *cattleyae*	0.63 ± 0.00 ^d^	20.82 ± 9.04 ^e^	500 ± 0.00 ^b^	0.07 ± 0.03 ^f^
*Acidovorax konjaci*	0.83 ± 0.36 ^d^	20.82 ± 9.04 ^e^	7.81± 0.00 ^e^	0.78 ± 0.00 ^de^
*Agrobacterium tumefaciens*	10.00 ± 0.00 ^a^	250.00 ± 0.00 ^a^	166.67 ± 72.17 ^c^	1.04 ± 0.45 ^cd^
*Burkholderia glumae*	10.00 ± 0.00 ^a^	250.00 ± 0.00 ^a^	20.84 ± 9.02 ^e^	0.1 ± 0.00 ^f^
*Clavibacter michiganensis* subsp. *michiganensis*	8.33 ± 2.29 ^a^	250.00 ± 0.00 ^a^	20.84 ± 9.02 ^e^	0.13 ± 0.06 ^f^
*Ewinia amylovora*	3.33 ± 1.44 ^bc^	62.50 ± 0.00 ^cd^	5.21 ± 2.25 ^e^	2.08 ± 0.91 ^ab^
*Pectobacterium carotovorum* subsp. *carotovorum*	2.50 ± 0.00 ^a^	52.08 ± 18.04 ^d^	15.63 ± 0.00 ^a^	1.56 ± 0.00 ^bc^
*Pectobacterium chrysanthemi*	3.33 ± 1.44 ^bc^	83.33 ± 36.08 ^c^	500 ± 0.00 ^b^	0.1± 0.00 ^f^
*Pseudomonas syringae* pv. *actinidiae*	2.50 ± 0.00 ^bc^	62.5 ± 0.00 ^cd^	15.63 ± 0.00 ^e^	2.61 ± 0.91 ^a^
*Pseudomonas syringae* pv. *lachrymans*	5.00 ± 0.00 ^b^	125.00 ± 0.00 ^b^	15.63 ± 0.00 ^e^	1.56± 0.00 ^bc^
*Ralstonia solanacearum*	1.67 ± 0.72 ^cd^	41.67 ± 18.04 ^de^	10.42 ± 4.52 ^e^	0.20 ± 0.00 ^ef^
Sr-Pcc ^4^	2.50 ± 0.00 ^cd^	62.5± 0.00 ^cd^	1000 ± 0.00 ^a^	1.56± 0.00 ^bc^
*Xanthomonas arboricola* pv. *pruni*	2.50 ± 0.00 ^cd^	62.5 ± 0.00 ^cd^	20.83 ± 9.02 ^e^	0.39 ± 0.00 ^ef^
*Xanthomonas axonopodis* pv. *citri*	3.33 ± 1.44 ^bc^	62.5 ± 0.00 ^cd^	62.5 ± 0.00 ^d^	0.78 ± 0.00 ^de^
*Xanthomonas euvesicatoria*	2.08 ± 0.72 ^cd^	52.08 ± 18.04 ^d^	31.25 ± 0.00 ^e^	0.13 ± 0.06 ^f^
*Xanthomonas oryzae* pv. *oryzae*	1.25 ± 0.15 ^cd^	41.67 ± 18.04 ^de^	10.42 ± 4.52 ^e^	0.78 ± 0.00 ^de^

**^1^** Data are from three independent triplicates and presented as mean ± standard deviation. Means within the same column denoted by the different letters are significantly (*p* < 0.05) different by Duncan’s multiple range test. The order was sorted in alphabetical order. **^2^** Ss: Streptomycin sulfate, **^3^** Oa: Oxolinic acid, **^4^** Sr-Pcc: streptomycin-resistant *P.*
*carotovorum* subsp. *carotovorum*.

**Table 2 microorganisms-10-00791-t002:** In vitro assay of antifungal activity of the fermentation filtrate (FF) and the partially purified fraction (PPF) of *Streptomyces* sp. AN090126 against various plant pathogenic fungi.

Plant Pathogenic Fungi	Minimum Inhibitory Concentration ^1^	
	FF (%)	PPF (µg/mL)
*Botrytis cinerea*	0.63 ± 0.00 ^c^	15.6 ± 0.00 ^d^
*Colletotrichum coccodes*	3.33 ±1.44 ^a^	83.33 ± 36.08 ^a^
*Cryphonectria parasitica*	1.67 ± 0.72 ^bc^	31.25 ± 0.00 ^cd^
*Fusarium graminearum*	1.25 ± 0.00 ^bc^	31.25 ± 0.00 ^cd^
*Fusarium oxysporum* f. sp. *lycopersici*	1.67 ± 0.72 ^bc^	52.08 ± 18.04 ^bc^
*Magnaporthe oryzae*	>10	>500
*Phytophthora cactorum*	1.25 ± 0.00 ^bc^	31.25 ± 0.00 ^cd^
*Raffaelea quercus-mongolicae*	2.08 ± 0.72 ^b^	52.08 ± 18.04 ^bc^
*Rhizoctonia solani*	1.04 ± 0.36 ^bc^	26.03 ± 9.04 ^cd^
*Sclerotinia homoeocarpa*	2.08 ± 0.72 ^b^	62.5 ± 0.00 ^ab^

**^1^** Data are from three independent triplicates and presented as mean ± standard deviation. Means within the same column denoted by the different letters are significantly (*p* < 0.05) different by Duncan’s multiple range test. The order was sorted in alphabetical order.

**Table 3 microorganisms-10-00791-t003:** Gas chromatography-mass spectrometry analysis of putative volatile organic compounds (VOCs) produced by *Streptomyces* sp. AN090126.

Putative VOC	Retention Time, min	Peak Area, % ^1^
2,4-Dimethyl hexanoic acid, methyl ester	2.94	6.16 ± 0.15 ^e^
3-Methyl-1-butanol	3.22	28.41 ± 0.52 ^a^
2-Methyl-1-butanol	3.27	17.24 ± 0.73 ^b^
Dimethyl disulfide	3.37	9.91 ± 0.37 ^d^
1-Methoxy-2-methyl benzene	3.68	2.94 ± 0.09 ^hi^
1-Hexanol	5.27	4.45 ± 0.13 ^f^
1,4-Dimethyl benzene	5.71	2.73 ± 0.16 ^i^
Dimethyl trisulfide	7.12	3.22 ± 0.32 ^hi^
2-Propyl furan	7.34	15.17 ± 0.16 ^c^
2-Ethyl-1-hexanol	8.06	3.40 ± 0.16 ^g^
1-Octanol	8.78	6.36 ± 0.13 ^e^

**^1^** Data are from three independent triplicates and presented as the mean ± standard deviation. Means within a column denoted by the different letters are significantly different (*p* < 0.05) by Duncan’s multiple range test. The order was sorted in increase in retention time.

**Table 4 microorganisms-10-00791-t004:** Interaction of the partially purified fraction (PPF) and streptomycin sulfate (Ss) against streptomycin-resistant bacteria using checkerboard assay.

Bacteria	FIC ^1^		FICI ^2^	Outcome
	Ss	PPF		
Aac ^3^	0.25 ± 0.00 ^b^	0.42 ± 0.14 ^a^	0.67 ± 0.14 ^ab^	Partial synergy
At	0.50 ± 0.00 ^a^	0.50 ± 0.00 ^a^	1.00 ± 0.00 ^a^	Additive
Pc	0.17 ± 0.07 ^c^	0.41 ± 0.14 ^a^	0.58 ± 0.19 ^b^	Partial synergy
Sr-Pcc	0.25 ± 0.00 ^b^	0.42 ± 0.14 ^a^	0.67 ± 0.07 ^ab^	Partial synergy
Xac	0.25 ± 0.00 ^b^	0.50 ± 0.00 ^a^	0.75 ± 0.00 ^ab^	Partial synergy

**^1^** FIC: fractional inhibitory concentration, **^2^** FICI: fractional inhibitory concentration index; ≤0.5, synergy; 0.5 < to ≤0.75, partial synergy; 0.75 < to ≤1; additive effect; 1 < to ≤4, indifference or no effect; and >4 antagonism., **^3^** Aac: *A. avenae* subsp. *cattleyae*; At: *A. tumefaciens*; Pc: *P. chrysanthemi*; Sr-Pcc: streptomycin-resistant *P. carotovorum* subsp. *carotovorum*; Xac: *X. axonopodis* pv. *citri*. Data are from three independent triplicates and presented as mean ± standard deviation. Means within the same column denoted by the different letters are significantly (*p* < 0.05) different according to Tukey’s HSD test.

## Data Availability

All data generated or analyzed during this study can be found in this published article.

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
