# Peer review of "Streptomyces sp. AN090126 as a Biocontrol Agent against Bacterial and Fungal Plant Diseases"

_microorganisms, 2022, doi:10.3390/microorganisms10040791_

Round 1

Reviewer 1 Report

Manuscript Title: Streptomyces sp. AN090126, as a biocontrol agent against bacterial and fungal plant diseases

Manuscript ID: microorganisms-1664522

The manuscript written by Le et al. reports very interesting results. In this work, the authors have discovered that Streptomyces sp. AN090126 exhibited potent antibacterial and antifungal activities as well as synergistic activity with streptomycin. Further research on this area may result in the development of Streptomyces sp. AN090126 for disease control.

In general, this paper is clearly laid out, well planed and easy to read. The experiments are well designed and appropriate controls are presented. Some specific suggestions or questions are listed below:
  1. Introduction: Introduction is easy to read but needs a little completed.Streptomyces strains are known for their ability to manage bacterial and fungal pathogens infecting crop plants. Authors should add more information into this section and cite the recent research into the field. I suggest the authors compare the antibacterial and antifungal activities of Streptomyces AN090126 with that of other Streptomyces strains based on the literature.
  2. Material and Methods(Line 81-82): Has Streptomyces AN090126 isolated in this study been deposited in a public strain collection? I suggest the authors add the collection number and mention it in the manuscript.
  3. Material and Methods(Line 194): GC-MS, please use full name for GC-MS for the first time.
  4. Material and Methods(Line 206): Please change “Plant material’ as “Plant materials”.
  5. Results (Line 264-274): Please ensure that abbreviations/acronyms are defined the first time they appear in each of three sections: the abstract; the main text; the first figure or table, then use the abbreviation.
  6. Discussion: The authors should compare the antibacterial and antifungal activitiesof Streptomyces AN090126 with that of other Streptomyces strains based on the literature.

Author Response

We truly appreciate the reviewer’s insightful comments and suggestions and express gratitude for the time spent reviewing our manuscript.

- Material and Methods(Line 81-82): Has StreptomycesAN090126 isolated in this study been deposited in a public strain collection? I suggest the authors add the collection number and mention it in the manuscript.

The AN090126 strain has not been deposited in the public bank yet. We are going to deposit the strain soon.

-  Material and Methods(Line 194): GC-MS, please use full name for GC-MS for the first time.

We use full name for GC-MS for first time in abstract section, please see L8.

-  Material and Methods(Line 206): Please change “Plant material’ as “Plant materials”.

We have changed “plant material” to “plant materials”, please see L178.

-  Results (Line 264-274): Please ensure that abbreviations/acronyms are defined the first time they appear in each of three sections: the abstract; the main text; the first figure or table, then use the abbreviation.

Accepted and deposited.

- Discussion: The authors should compare the antibacterial and antifungal activities of Streptomyces AN090126 with that of other Streptomyces strains based on the literature.

Accepted and deposited. Please see L382-387.

Reviewer 2 Report

Report about: Streptomyces sp. AN090126, as a biocontrol agent against bacterial and fungal plant diseases microorganisms-1664522.

The topic of the paper is appropriated for the journal.

The title of the manuscript reflected the content of the paper.

The aims of the study are clearly stated.

The paper is well written and easy to follow.

This paper can be accepted in the journal but needs minor revision. This is an excellent paper but many references needs to be added in addition to add some parts in the introduction about the importance of actinobacteria as biocontrol agent and as plant growth promotors.

  • Tables and figures are NOTs self-explanatory.
  • Please mention the number of replicates in all tests.
  • Please do not write spp. in italic format.
  • For the scientific names, only you write it in full at the first mention. Then later on, you need to use the first letter only of the genus name.
  • Remarks
  • The paper is missing many references that must be added.

10.3389/fmars.2021.710200 and 10.1007/s00374-020-01450-3

10.3389/fmicb.2018.00829 and 10.3389/fmicb.2017.01455

10.3389/fmicb.2020.00552 and 10.3389/fmicb.2019.01694

https://doi.org/10.1016/j.biocontrol.2021.104783 and this 10.3390/jof7110885

doi: https://doi.org/10.3390/jof8010008

  • Please provide the full details of the company, city name, country for all the chemicals purchased in this study.
  • Please include the city, country for all the machines and companies that you used in the entire paper.
  • Please write at the first mention in FULL what do you mean by PAGE and PUP/NUP and SDS??
  • Please make all tables and figures MUST be self-explanatory.
  • Please check the scientific names in the references list as some of them are NOT written in italic format.
  •  

So my final recommendations is to accept the paper with minor revision.

I must revise the paper again to check that my comments were all done.

Author Response

We truly appreciate the reviewer’s insightful comments and suggestions and express gratitude for the time spent reviewing our manuscript.

-  Tables and figures are NOTs self-explanatory.

Accepted and deposited.

-  Please mention the number of replicates in all tests.

We have provided the number of replicates in all figures and tables.

-  Please do not write spp. in italic format.

Accepted and deposited. Please see L34, 36, 45, 50, 55, 84, 225, 367, 394, 398, 413, 436, 518, and 562.

-  For the scientific names, only you write it in full at the first mention. Then later on, you need to use the first letter only of the genus name.

Accepted and deposited.  

-  Remarks

The paper is missing many references that must be added.

10.3389/fmars.2021.710200 and 10.1007/s00374-020-01450-3

10.3389/fmicb.2018.00829 and 10.3389/fmicb.2017.01455

10.3389/fmicb.2020.00552 and 10.3389/fmicb.2019.01694

https://doi.org/10.1016/j.biocontrol.2021.104783 and this 10.3390/jof7110885

doi: https://doi.org/10.3390/jof8010008

Accepted and deposited. Please see We have added these references into Reference section, please see L520-528, 535-542, 584-588, 594-603, and 607-610.

- Please provide the full details of the company, city name, country for all the chemicals purchased in this study.

Accepted and deposited. Please see L68-70, 75-76, 87-88, 96-97, 113, 114-115, 120, 123-124, 132-133, 179-181, and 195-197.

- Please include the city, country for all the machines and companies that you used in the entire paper.

Accepted and deposited. Please see L114-115, 162-163, and 171-172.

- Please write at the first mention in FULL what do you mean by PAGE and PUP/NUP and SDS??

Accepted and deposited.

- Please make all tables and figures MUST be self-explanatory.

Accepted and deposited.

- Please check the scientific names in the references list as some of them are NOT written in italic format.

Accepted and deposited. Please see L530, 554, 590, 592, and 644.

Reviewer 3 Report

Manuscript ID microorganisms-1664522 "Streptomyces sp. AN090126, as a biocontrol agent against bacterial and fungal plant diseases" by Duy Le and collaborators explores the potential of partial purified fraction and volatile organic compounds of fermentation broth from Streptomyces. The work is nicely presented with appropriate experimental design and statistical analyses. The Introduction contains appropriate level of detail and results were discussed in light of the available literature. Although the exact compounds responsible for the antimicrobial effect were not identified, the authors did provide some suggestions that could be further investigated. The research has potential for the development of novel formulations that could substitute or complement synthetic treatments currently available. I believe the manuscript is of enough interest to readers of Microorganisms and is ready for publication after minor suggestions I listed below.

Minor suggestions:
line 28: remove "as"
line 36: consider replacing "In this light" for "In light of this"
line 87: confirm the pH is +- 2. This seems a wide range (pH 5-9)
line 144: review strange character between "4" and "for 15 mins"
lines 238-240: since SEM images are grayscale, consider rewording this sentence for clarity. The greenish gray color might have been observed on the sample that was then used for SEM.
line 253: "different genome species"
line 417: remove the double space between "euvesicatoria" and "Agrobacterium"
Figures and Tables legends: Define all abbreviations, even if used previously in the main text. All figures and legends should be self-explanatory without the main text.

Author Response

We truly appreciate the reviewer’s insightful comments and suggestions and express gratitude for the time spent reviewing our manuscript.

- Line 28: remove "as"

Accepted and deposited. Please see L17.

- Line 36: consider replacing "In this light" for "In light of this"

Accepted and deposited. Please see L24.

- Line 87: confirm the pH is +- 2. This seems a wide range (pH 5-9)

Actually, the pH is 7,0 ± 0.2. To reduce the DUPLICATION, we revised the sentence. Please see L72.

- Line 144: review strange character between "4" and "for 15 mins"

We revised them. Please see L120-121.

- Lines 238-240: since SEM images are grayscale, consider rewording this sentence for clarity. The greenish gray color might have been observed on the sample that was then used for SEM.

Accepted and deposited. Please see L210.

- Line 253: "different genome species"

Accepted and deposited. Please see L224-225.

- Line 417: remove the double space between "euvesicatoria" and "Agrobacterium"

Accepted and deposited. Please see L404.

- Figures and Tables legends: Define all abbreviations, even if used previously in the main text. All figures and legends should be self-explanatory without the main text.

Accepted and deposited.

Reviewer 4 Report

The manuscript presents the results obtained using metabolites produced by a strain of Streptomyces sp. (Streptomyces sp. AN090126) isolated from agricultural suppressive soil of Korea, against phytopathogenic microorganisms (bacteria and fungi).  The isolated strain was characterized by molecular analysis.  Submerged biosynthesis was made with this strain, and the fermentation broth obtained was fractionated and tested, ''in vitro'' and ''in vivo'' compared with  Streptomycin sulfate and  Oxolinic acid. 
The article is well written and even it contains a lot of information,  these are presented concisely. Results obtained are correctly presented;  discussion is conducted logically and was made in comparison of stat of arts in the field. 
 The results obtained from studies performed ''in vitro'' and ''in vivo'' propose to use the strain Streptomyces sp. AN090126 is a potential biocontrol agent for the control of plant diseases caused by bacteria and fungi.  This proposal is feasible because the authors demonstrated the ability of Streptomyces sp. AN090126  to produce water-soluble secondary antimicrobial metabolites and volatile organic compounds with broad-spectrum antimicrobial activity. For these reasons, I propose article publishing, in this form.

Author Response

We truly appreciate the reviewer’s insightful comments and suggestions and express gratitude for the time spent reviewing our manuscript. 

Round 2

Reviewer 2 Report

This paper can NOW be accepted. The authors did all my corrections. Well Done.